# p21 as a Predictor and Prognostic Indicator of Clinical Outcome in Rectal Cancer Patients

**DOI:** 10.3390/ijms25020725

**Published:** 2024-01-05

**Authors:** Li Ching Ooi, Vincent Ho, Jing Zhou Zhu, Stephanie Lim, Liping Chung, Askar Abubakar, Tristan Rutland, Wei Chua, Weng Ng, Mark Lee, Matthew Morgan, Scott MacKenzie, Cheok Soon Lee

**Affiliations:** 1Department of Anatomical Pathology, Liverpool Hospital, Liverpool, NSW 2170, Australia; liching.ooi8@gmail.com (L.C.O.); zhujingzhou@gmail.com (J.Z.Z.); t.rutland@westernsydney.edu.au (T.R.); soon.lee@westernsydney.edu.au (C.S.L.); 2School of Medicine, Western Sydney University, Penrith, NSW 2751, Australia; liping.chung@westernsydney.edu.au (L.C.); askar.abubakar@westernsydney.edu.au (A.A.); wei.chua@health.nsw.gov.au (W.C.); s.mackenzie@westernsydney.edu.au (S.M.); 3Ingham Institute for Applied Medical Research, Liverpool, NSW 2170, Australia; stephanie.lim@health.nsw.gov.au; 4Macarthur Cancer Therapy Centre, Campbelltown Hospital, Campbelltown, NSW 2560, Australia; 5Discipline of Medical Oncology, School of Medicine, Western Sydney University, Liverpool, NSW 2170, Australia; 6Discipline of Pathology, School of Medicine, Western Sydney University, Campbelltown, NSW 2560, Australia; 7Department of Medical Oncology, Liverpool Hospital, Liverpool, NSW 2170, Australia; weng.ng@health.nsw.gov.au; 8Department of Radiation Oncology, Liverpool Hospital, Liverpool, NSW 2170, Australia; mark.lee2@health.nsw.gov.au; 9Department of Colorectal Surgery, Liverpool Hospital, Liverpool, NSW 2170, Australia; mjmorgan@bigpond.com

**Keywords:** p21, rectal cancer (CRC), tissue microarrays, radiotherapy

## Abstract

The cell cycle plays a key and complex role in the development of human cancers. p21 is a potent cyclin-dependent kinase inhibitor (CDKI) involved in the promotion of cell cycle arrest and the regulation of cellular senescence. Altered p21 expression in rectal cancer cells may affect tumor cells’ behavior and resistance to neoadjuvant and adjuvant therapy. Our study aimed to ascertain the relationship between the differential expression of p21 in rectal cancer and patient survival outcomes. Using tissue microarrays, 266 rectal cancer specimens were immunohistochemically stained for p21. The expression patterns were scored separately in cancer cells retrieved from the center and the periphery of the tumor; compared with clinicopathological data, tumor regression grade (TRG), disease-free, and overall survival. Negative p21 expression in tumor periphery cells was significantly associated with longer overall survival upon the univariate (*p* = 0.001) and multivariable analysis (*p* = 0.003, HR = 2.068). Negative p21 expression in tumor periphery cells was also associated with longer disease-free survival in the multivariable analysis (*p* = 0.040, HR = 1.769). Longer overall survival times also correlated with lower tumor grades (*p*= 0.011), the absence of vascular and perineural invasion (*p* = 0.001; *p* < 0.005), the absence of metastases (*p* < 0.005), and adjuvant treatment (*p* = 0.009). p21 expression is a potential predictive and prognostic biomarker for clinical outcomes in rectal cancer patients. Negative p21 expression in tumor periphery cells demonstrated significant association with longer overall survival and disease-free survival. Larger prospective studies are warranted to investigate the ability of p21 to identify rectal cancer patients who will benefit from neoadjuvant and adjuvant therapy.

## 1. Introduction

Colorectal cancer (CRC) is the third most common cancer in Western countries, with one of the highest rates seen in Australasia. About one third of colorectal tumors arise from the rectum [1]. In contrast to colonic cancer patients, patients with rectal tumors often present at a locally advanced stage, have poorer clinical outcomes, and a poorer quality of life [1]. The current management for primary rectal adenocarcinoma is largely multimodal, involving a combination of radiotherapy, chemotherapy, and surgery. This is particularly pertinent to patients with Stage II to IV rectal cancer, wherein survival rates range from approximately 25% to 90% [2]. Hence, it is essential for us to ascertain biomarkers that can best predict tumor prognosis, inform the choice of therapy, evaluate treatment response, and aid risk stratification.

p21 is a crucial protein which inhibits cell cycle progression primarily through binding and inhibiting several cyclins and cyclin-dependent kinases, such as the CDK2, CDK1, and CDK4/6 complexes [3]. It has additional key roles in apoptosis, the reprogramming of induced pluripotent stem cells, differentiation, transcription, DNA repair, and cell migration [4,5].

p21 is conventionally known for its tumor suppressor role of causing cell cycle arrest through its p53-dependent and p53-independent pathways [6,7,8]. p21 is downstream of P53 and considered a master regulator of senescence [9]. p21 is also involved in DNA repair and synthesis [10]. There is emerging evidence suggesting that p21 is a potential oncogene by protecting cells from apoptosis. We therefore hypothesize that p21 expression deficiencies in rectal cancer cells may lead to more indolent tumor behavior, hence correlating with better clinical outcomes.

The functional significance of p21 in tumorigenesis is not clearly understood; however, many studies have found the overexpression of p21 levels in a range of human cancers including cervical, breast, and prostate cancer [11,12,13,14,15,16]. The tumor suppressor role of p21 is supported by evidence from multiple studies of CRC patients and several studies of rectal cancer patients [17,18,19,20,21,22,23,24]. However, as an apoptosis modulator, p21 may assume an oncogenic role through its involvement with apoptosis-inducing proteins [4]. This is consistent with a study by Noske et al. [25] that showed that positive p21 expression in CRC patients post adjuvant treatment correlated with poorer prognosis and is consistent with multiple studies of rectal cancer patient cohorts. Both Sturm et al. [26] and Rau et al. [27] found that patients with higher p21-expressing cancer cells post neoadjuvant or adjuvant treatment experienced poorer survival outcomes. It is suspected that without intact cell cycle checkpoints in p21-deficient tumor cells, the DNA repair process mediated by p21 does not occur, minimizing the likelihood of developing resistance [28]. Additionally, the cancer cells easily proceed through to the S phase of cell cycle, accumulating multiple replication errors with uncontrolled mitosis, leading to eventual apoptosis especially under the stress of ionizing or chemotherapeutic agents [29,30,31].

These findings suggest that p21 could provide a promising predictive and prognostic biomarker, which would contribute to more personalized patient-centered care, enhance healthcare quality, and avoid overtreatment.

## 2. Results

### 2.1. Patient Characteristics of Entire Cohort

Two hundred and sixty-six cases were identified, with a mean age of 70.9 years (range 35–100) (Table 1). The cohort consisted of 66.2% males, 33.8% females, 33.5% pT1/2, 66.5% pT3/4, 46.3% node-positive, and 7.1% metastatic disease. Twenty-two percent received neoadjuvant therapy, whilst 30.8% received adjuvant therapy.

The DFS and OS data were available for 199 and 249 patients, respectively. Tumor recurrence (either local or distant) occurred in 83 patients. At the time of the study, 142 of 251 patients (56.6%) were alive. The averages of both the disease-free survival and overall survival were 3.8 years (range 0–12.60). The median time to recurrence and death were 2.78 years and 3.19 years, respectively.

### 2.2. Patient Characteristics of Subcohort Who Received Neoadjuvant Therapy

Amongst the fifty-four patients who received neoadjuvant treatment, local recurrence occurred in 46.2% of the cohort, and 59.3% were alive at the time of data collection. The mean recurrence was 3.2 years (range 0.1–11.8) post-surgery despite neoadjuvant therapy, whilst the mean time to recurrence was 2.25 years. The median overall survival time was 3.9 years (range 0.17–11.81), whilst the median overall survival time was 3.08 years (Table 2).

### 2.3. p21 Expression

The p21 expression was analyzed in 265 TC and 263 TP samples (Table 3). The areas with the highest mitotic activity in the central region of the cancer were designated as the TC, whereas the most mitotically active areas at the outer invasive zone of the tumor were considered the TP. Among the TC samples, negative p21 expression was seen in 199/265 (75.1%) and positive expression in 66/265 (24.9%) cases. In the TP cases, negative expression was seen in 171/263 (65%) and positive expression in 92/263 (35%) cases.

In our dataset, p21 positive expression is significantly greater in the TP samples compared to the TC samples (*p* = 0.013 on Fisher’s exact test).

The clinicopathological and clinical outcome data were analyzed separately for the TC and TP p21 expression.

A total of 115 lymph nodes in node-positive cases were stained. We found that 94/115 (81.7%) lymph node-positive samples had negative p21 expression, whereas 21/115 (18.3%) lymph node-positive specimens were positive for p21 expression.

In regard to the samples of normal rectal tissue, the p21 staining results were available for 248 NCT and 256 NAT cases. p21 expression was positive in 156/248 (62.9%) NCT samples and in 193/256 (75.4%) NAT samples.

### 2.4. Association of p21 Expression with Clinicopathological Variables

The p21 expression in the TC samples was significantly associated with the nodal status (x^2^(1) > 3.989; N = 115; *p =* 0.046) and tumor recurrence status (x^2^(1) > 4.767; N = 258; *p =* 0.029) (Table 4). There were no other significant correlations within the TC samples. The expression of p21 in the TP cases was not associated with any clinicopathological variables.

### 2.5. Association of p21 Expression with Disease-Free Survival and Overall Survival in Entire Rectal Cancer Cohort

The DFS and OS outcomes were analyzed in 199 and 249 patients, respectively (Table 5 and Table 6).

Longer OS outcomes were significantly associated with negative p21 expression in the TP samples (*p =* 0.001; Figure 1). After adjusting for confounders in the multivariable analysis (Table 6), the association between the OS and negative p21 expression in the TP tissues remained significant, independent of the presence of metastases, perineural invasion, or of adjuvant therapy (p21, TP-available cases [HR = 2.068 (1.290–3.316), *p =* 0.003]).

There was initially no significant association between the DFS and p21 expression in the TP cases on the univariate analysis (*p =* 0.152; Figure 2). However, upon modifying for confounders in the multivariate Cox regression analysis, negative p21 expression in the TP samples correlated with a longer DFS, independent of perineural invasion or being recipients of neoadjuvant treatment (p21, TP-available cases [HR = 1.769 (1.027–3.049), *p =* 0.040]). This indicates that whilst Kaplan–Meier curves can provide informative visualization, there are distinct limitations, such as the inability to control for confounding variables. Cox regression, on the other hand, provides a more comprehensive analysis by accounting for covariates.

The p21 expression in the TC samples did not correlate significantly with the OS (*p =* 0.843) or DFS outcomes (*p =* 0.149), even after the multivariable analysis for the OS (p21, TC-available cases [HR = 0.888 (0.503–1.566), *p =* 0.682]) or DFS (p21, TC-available cases [HR = 1.422 (0.737–2.745), *p =* 0.294]).

As expected, larger tumor sizes and the presence of vascular or perineural invasion were associated with shorter OS outcomes (*p =* 0.011; *p =* 0.001; *p* < 0.0001, respectively) and a worse DFS (*p =* 0.001; *p =* 0.041; *p* < 0.0001, respectively). The patients who did not have metastases and who received adjuvant therapy exhibited longer overall survival times (*p* < 0.0001; *p =* 0.009, respectively). The rectal cancer patients with higher grades and the recipients of neoadjuvant therapy also had a worse DFS (*p =* 0.037; *p =* 0.021, respectively).

The presence of perineural invasion was shown to negatively impact the OS (p21, TP-available cases [HR = 2.184 (1.194–3.996), *p =* 0.011] and p21, TC-available cases [HR = 1.951 (1.075–3.541), *p =* 0.028]) and DFS (p21, TP-available cases [HR = 2.465 (1.312–4.631), *p =* 0.005] and p21, TC-available cases [HR = 2.182 (1.188–4.006), *p =* 0.012]).

As expected, the patients without metastases had a better OS (p21, TP-available cases [HR = 3.444 (1.422–8.342), *p =* 0.006] and p21, TC-available cases [HR = 3.617 (1.517–8.621), *p =* 0.004]), as did those patients who were recipients of adjuvant therapy (p21, TP-available cases [HR = 0.346 (0.196–0.613), *p* < 0.0001 and p21, TC-available cases [HR = 0.332 (0.186–0.590), *p* < 0.0001]).

The cohort of locally advanced rectal cancer (LARC) patients who received neoadjuvant treatment had worse DFS outcomes (p21, TP-available cases [HR = 1.816 (1.034–3.191), *p =* 0.038] and p21, TC-available cases [HR = 1.875 (1.062–3.311), *p =* 0.030]).

### 2.6. Association of P21 Expression with Neoadjuvant Treatment and Response

There was no significant correlation between the p21 expression in either the TP or TC samples and neoadjuvant treatment in the LARC subgroup (Table 7).

TRG data were obtained for 51/54 (94.4%) patients who received pre-operative therapy. Grades 0–1 were considered a “good response”; grades 2–3 were considered a “poor response”. Out of this patient cohort, forty-five patients (88.2%) demonstrated poor response whilst only six patients (11.8%) showed a good response. The number of good responders was too low for meaningful statistical analysis.

## 3. Discussion

We conducted this study to determine the expression pattern of p21 in rectal cancer cells and the implication of p21 expression on rectal cancer progression.

Regardless of its origin site, our results showed that the majority of the tumor tissues (TC, TP, and LN) did not stain positively for p21, whereas the normal mucosal tissues had positive p21 expression (NAT and NCT). This is in line with previous studies showing an inverse relationship between the p21 expression and proliferation, with terminally differentiated cells generally showing p21 positivity [32,33]. In light of the variable p21 expression between the TC and TP, this may represent the heterogeneity in the cellular biology between the central and peripheral cancer cells within the same rectal tumor. The analysis of p21 expression in TP cells is predicted to better reflect the invasiveness and aggressive nature of rectal tumors because of their better vascular supply and interaction with surrounding tissues. Whilst the tumor center is useful, cells within the tumor center are generally necrotic, ischemic, and exhibit restricted growth due to their limited vascular supply [34].

p21 is a known direct transcriptional target of tumor suppressor p53. Our study revealed that the positive p21 expression in the TC samples was significantly associated with a negative nodal status. Additionally, there was also significant correlation between positive p21 expression in the TC cells with no recurrence. Our results support the anti-proliferative role of p21 in the cell cycle, mediating the cell cycle arrest process in cells. However, a different picture is presented when p21 expression is analyzed against clinical outcomes.

Our main finding is that negative p21 expression in the TP tissues was linked to better overall survival, independent of the tumor’s perineural invasion status, metastases status, or whether patients received adjuvant therapy.

There are a few reasons that may explain this finding. The mean age in our cohort is 70.9 years. Negative p21 expression has been associated with a longer survival in colon cancer patients for those 60 years and over, and a shorter survival among patients less than 60 years of age [35]. In the non-neoplastic state, the function of p21 has been related with the cellular senescence and aging of stem cells [36].

Cancerous stem cells which have been close to senescence or in the state of senescence (in old individuals) may be more susceptible to the apoptotic signal when the cell cycle is not blocked by p21. In contrast, in cancerous stem cells in young individuals, the adverse effect of cell cycle progression by p21 loss may have a more direct influence on tumor behavior. Therefore, it is entirely possible that stem cells that give rise to tumors in older individuals may have substantially different molecular features from stem cells that give rise to tumors in younger individuals. P21 loss thus could be a marker for aggressive tumors in a subset of younger persons, and a marker for a less-aggressive tumor in a subset of older persons, in the context of a particular host microenvironment.

In our study, we found p21 positive expression to be significantly greater in the tumor periphery versus the tumor core. This mirrors research into lung adenocarcinoma where positive 21 expression was predominant in the tumor periphery [37].

In the study of lung adenocarcinoma, p21 expression was found to be moderately up-regulated in the tumor periphery, and functions to promote cyclin A-cdk2 assembly and kinase activity. It is known that for both human lung adenocarcinoma [38] and colon cancer [39], the elevated expression of active cyclin A-CDK2 complexes with associated higher CDK2 kinase activity is critical in the promotion of cell cycle progression and the unrestrained proliferation of tumor cells, thereby being a predictive marker for patients’ prognosis.

It stands to reason, therefore, that the converse, i.e., the negative expression of p21 in the tumor periphery may lead to more favorable clinical outcomes.

Indeed, in our study, p21 expression also significantly correlated with a better DFS only after multivariable analysis, independent of the tumor category and neoadjuvant therapy. Recent studies have questioned p21 as an oncogene, particularly with regard to apoptosis induced by DNA damage [4,26,40,41]. Several in vitro studies have demonstrated that increased p21 expression is associated with greater cellular arrest, conferring greater resistance against apoptosis in human carcinoma cells under the stress of DNA damage [41,42]. Our results are consistent with pre-existing studies that linked a higher p21 expression with poorer clinical outcomes [43,44]. Reerink et al. [43] and Sim et al. [44] analyzed the p21 levels in rectal cancer specimens prior to any treatment including chemoradiation, concluding that positive p21 expression correlated with poor pathological responses, poorer prognosis, and worse survival rate. Nevertheless, most past studies have focused on rectal cancer patients who were treated with either chemotherapy, radiotherapy, or a combination. Our data here reveal a significant impact on the clinical outcome independent of adjuvant or neoadjuvant therapies. This could possibly suggest additional factors at play beyond the ones suggested.

In contradistinction, our results were inconsistent with other studies that linked positive p21 expression with better outcomes [21,22,45,46]. The two studies conducted by Suzuki et al. [45] and Charara et al. [22] remarked that positive p21 levels in post-CRT rectal cancer tissues correlated with higher histological regression and better pathological response to regimens. Findings by Schwandner et al. [46] also concluded that p21 is an independent prognostic predictor in rectal cancer for DFS and recurrence. Interestingly, a study by Sturm et al. [6] found that patients with positive pre-therapeutic p21 expression had better local tumor responses; however, the persistence of such levels four to six weeks post completion of neoadjuvant CRT was linked to poorer survival. It was hypothesized that this reflected a potential selection process for tumor-resistant cells due to the role of p21 in facilitating DNA repair, hence repairing cancer cells affected by the DNA damage response [26]. There are also many studies showing insignificant results between p21 expression and clinical outcomes [29,47,48]. This can be explained if we understand p21 as not simply a tumor suppressor but also as possessing oncogenic potential. P21′s oncogenic potential has been attributed to its cytoplasmic localization [49], the promotion of cell cycle progression, the favoring of migration [5], and the inhibition of apoptosis [50]. Similar to proteins such as MTDH [51], it is able to modulate multiple oncogenic pathways. Additionally, p21′s sustained overexpression can lead to bypassing/escaping from senescent cell arrest [52]. Its dual behavior in various processes dependent on the cellular and environmental context can often lead to opposing cellular responses [53].

Our data demonstrated that the patients with lower-grade tumors, lower TNM staging, and tumors without vascular and peri-neural invasion had a longer DFS. Additionally, the subgroup of patients with LARC who received neoadjuvant treatment were shown to have worse DFS outcomes. This appears to reflect an intrinsically poorer prognosis for the patient subpopulation selected for neoadjuvant therapy.

The main limitation of our study is that because it is a retrospective study, there is incomplete data available on treatment regimens and the cause of death. Therefore, we were unable to determine whether the cause of death was rectal cancer-specific. Furthermore, there was incomplete data regarding the specified type of neoadjuvant and adjuvant treatment received by patients.

In regard to incomplete data, we are cognizant of the multiple efforts [54] to develop imputation algorithms for guiding missing value imputation, which include k-nearest neighbors imputation (KNN), random forest (RF), singular value decomposition-based imputation (SVD), and Bayesian principal component analysis (BPCA). We did not utilize these techniques but acknowledge that even with a very strict correction technique as suggested by Bonferroni [55], this may lead to the preservation of little statistical power and consequently may result in no significant findings.

The neoadjuvant subcohort was small, consisting of only 54 patients. This small cohort size makes it difficult to assign significance to p21 expression as a biomarker for resistance to neoadjuvant therapy in the rectal cancer group. Analysis from ongoing, larger-scale trials will be needed to definitively answer the pertinent questions raised in this report.

Lastly, there is no widely recognized cut-off threshold for p21 expression; hence, this may have contributed to inconsistencies in results across previous studies.

In conclusion, our study showed that the p21 expression was lower in the rectal tumor samples compared to the normal mucosal tissues. Furthermore, the expression of p21 within the tumor itself was heterogenous, with more TC cells displaying negative p21 expression than TP cells. Interestingly, negative p21 expression in the TC cells significantly correlated with positive nodal status and recurrence. This is in line with the known anti-proliferative role of p21. However, upon multivariable analysis, our most pertinent finding was that the negative p21 expression in the TP cells was associated with better overall survival and disease-free survival outcomes, independent of confounding factors. This was consistent with our hypothesis and suggests that p21 expression may confer more aggressive rectal cancer behavior.

Larger prospective studies are needed to investigate the utility of p21 as a biomarker to identify rectal cancer patients who will benefit from neoadjuvant and adjuvant therapy. A greater understanding of the biological role of p21 in rectal cancer may pave the way for other novel targeted therapies such as immunotherapy and chemodynamic therapy [56]. In light of emerging evidence, p21 remains a promising candidate as a biomarker of clinical outcome in rectal cancer; and may assist in identifying patients who will require more aggressive treatment.

## 4. Materials and Methods

### 4.1. Patient Samples

This project was approved by the South Western Sydney Local Health District (SWLHD) Human Research Ethics Committee (HREC/12/LPOOL/12). There were 266 specimens obtained from rectal cancer patients who underwent surgery in SWLHD between 2000 and 2011.

The histopathological and clinical outcomes data were obtained from three main databases—Powerchart, MOSAIQ, and Clinical Cancer Registry, Australia. The clinicopathological data are summarized in Table 8.

Variables of interest included age, sex, tumor staging based on the American Joint Committee on Cancer (AJCC) tumor-node-metastases (TNM) system, grade, vascular invasion, perineural invasion, and treatment. Outcomes of interest were overall survival (OS) and disease-free survival (DFS) for the entire cohort, and tumor regression grade (TRG) for cases treated with neoadjuvant therapy. OS was measured from the time of surgery or pathological diagnosis until time of death or to last known follow-up date. Insufficient information was available to assign exact cause of death. DFS was calculated from the time of surgery or pathological diagnosis, until the first appearance of either local or distant recurrence pathologically or radiologically. TRG was graded histologically according to the 7th edition of the AJCC cancer staging manual [57]. Our study dichotomized TRG as “good response” represented by grades 0 to 1, or “poor response” represented by grades 2 and 3.

### 4.2. Tissue Microarrays

The tissue microarray (TMA) was constructed from the resection specimens of primary rectal cancer tumors of 265 patients. The patients were recruited on the basis of the following criteria: histologically confirmed adenocarcinoma of the rectum; surgical resection of primary rectal cancer; age ≥ 18 years; and those who underwent surgery in South Western Sydney Local Health District (SWSLHD) between January 2000 and December 2011. We identified 266 patients.

Histopathology slides were scanned at 20× magnification using an Aperio ScanScope Model CS digital scanner. In order to avoid manual selection bias [58] cores with less than 30% tissue present or less than 100 cells were discarded.

Tissue microarrays (TMAs) were prepared from formalin-fixed, paraffin wax-embedded rectal cancer specimens. Using histological slides as references, tissue cores of 0.6 mm diameter were obtained from the normal tissue away from the tumor (NAT), normal tissue close to the tumor (NCT), tumor center (TC), tumor periphery (TP), lymph nodes (LN) in node-positive cases, hyperplastic polyp (P), and adenoma (Ade) if available. TC referred to the areas with highest mitotic activity at the center of the rectal cancer, whilst TP referred to the most mitotically active areas at the outer invasive zone of the tumor. Along with appropriate controls, the tissue cores were inserted into a paraffin block and sectioned into slides. After immunohistochemistry, the sections were analyzed for p21 expression.

### 4.3. Immunohistochemistry

Slides were pre-heated in an oven at 60 °C for 90 to 120 min. They were then deparaffinized in xylene and rehydrated in graded ethanol. Antigen retrieval was performed with Envision^TM^ FLEX Target Retrieval Solution High pH DM828 for 45 min in a 98 °C water bath. The slides were then cooled within the antigen retrieval buffer until the temperature reached 65 °C.

Slides were transferred to a Coplin jar filled with Envision™ FLEX Wash Buffer DM831 and incubated for 5 min at room temperature with Envision™ FLEX Peroxidase-Blocking Reagent DM821 to block endogenous peroxidases. Slides were then washed twice with Tris-Buffered Saline with Tween 20 (TBST) and incubated for 60 min with monoclonal anti-p21 primary antibody (1:25). After washing with TBST, the Dako mouse linker (Dako, North Sydney, Australia) was added to the slides, and incubated for 30 min with anti-mouse secondary antibody. After washing, peroxidase substrate (a combination of Envision™ FLEX DAB + Chromogen DM827 and Envision™ FLEX Substrate Buffer DM823) was used to develop color. Slides were counterstained with haematoxylin, washed with cold water, dipped 20 times in Scott Bluing solution and left to dry at room temperature.

### 4.4. Immunohistochemical Scoring

p21 expression was initially scored as the percentage of nuclear staining within each core in five percent increments. In light of recommendations from previous published TMA methods, we used a two-category classification for p21 scoring [59]. Results were assessed based on the estimated percentage of positive cells: 0 = 0%; 1 = <5%; 2 = 5–50%; 3 = 50–90%; and 4 = >90% using a semiquantitative score. Our threshold for positive p21 expression was 5% and greater (score 2–4); whilst p21 staining was considered to be negative when the percentage was lower than 5% (score 0–1). Since there were two duplicate punches for TC and TP variables, average protein expression was calculated before samples were dichotomized into positive (≥5%) or negative (<5%). We did not assess the intensity of immunohistochemical staining.

All immunohistochemically stained slides (Figure 3 shows an example) were independently assessed, with the observer blinded to clinical and pathological data. Prior to scoring, observers received standardized training from a trained pathologist. A second independent observer graded randomly selected 10% of cases. The scores were compared with each other to ensure consistency of scoring between the two independent observers. Interobserver variability was noted to be less than 1%.

### 4.5. Statistical Analysis

The collated clinicopathological data was entered and analyzed using IBM SPSS Statistics for Macintosh, Version 22.0 (IBM Corp. 2013. Armonk, NY, USA). The data collected were shown to be non-parametric. It should be noted that there is no standardized method for setting the cut-off value for the categorization of immunopositivity. However, maximally selected chi square statistics are traditionally employed here in a systematic fashion [60] and we use the chi-square test to analyze the differences between p21 expression of tissue types and various clinicopathological categories.

The survival analyses of OS and DFS were performed with the Kaplan–Meier method and log-rank test. The Kaplan–Meier method was used to ascertain the OS and DFS and has the advantage of considering time and censoring of data, which is lacking with receiver operator curve (ROC) analyses. The log-rank test was used to test deviation from the null hypothesis.

Univariate and multivariable analyses were performed using Cox proportional hazards survival modelling for p21 expression from TC and TP samples; covariates were sex, age, pathological TNM staging of tumor, grade, vascular invasion, perineural invasion, treatment received, and TRG. This analysis evaluated the independent effect of p21 status on mortality and recurrence. *p* < 0.05 is considered to be statistically significant.

## Figures and Tables

**Figure 1 ijms-25-00725-f001:**
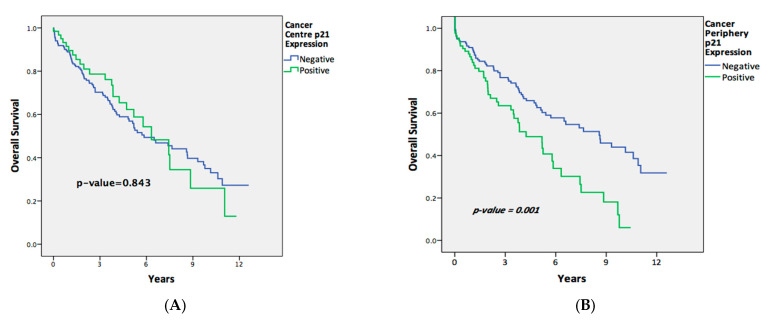
Overall survival of patients comparing p21-positive and p21-negative expression in (**A**) tumor center tissues and (**B**) tumor peripheral tissues.

**Figure 2 ijms-25-00725-f002:**
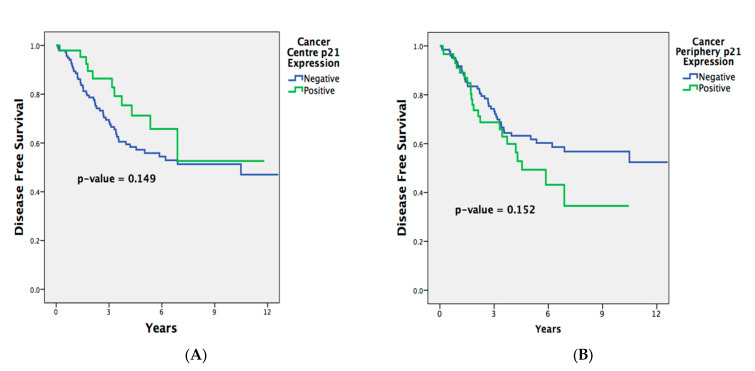
Disease-free survival of patients comparing p21-positive and p21-negative expression in (**A**) tumor center tissues and (**B**) tumor periphery tissues.

**Figure 3 ijms-25-00725-f003:**
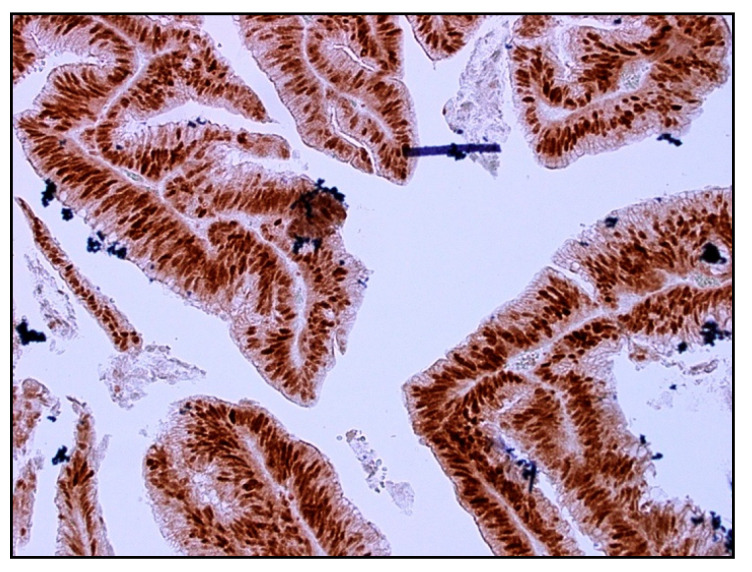
Positive nuclear immunoreactivity for p21 in rectal adenocarcinoma (Dako, anti-p21 primary antibody [1:25]), 20× magnification.

**Table 1 ijms-25-00725-t001:** Patient characteristics of entire cohort.

	All Patients
Total, n	266
Sex, n (%)	
Male	176 (66.2)
Female	90 (33.8)
Mean age, yrs	70.9
pT category, n (%)	
T1–2	87/260 (33.5)
T3–4	173/260 (66.5)
pN category, n (%)	
N0	139/259 (53.7)
N1–2	120/259 (46.3)
pM category, n (%)	
M0	223/240 (92.9)
M1	17/240 (7.1)
Grade, n (%)	
1–2	246/266 (92.5)
3	20/266 (7.5)
Vascular invasion,	
n (%)	
Absent	201/263 (76.4)
Present	62/263 (23.6)
Perineural invasion,	
n (%)	
Absent	220/263 (83.7)
Present	43/263 (16.3)
Neoadjuvant Treatment, n (%)	
Negative	191/245 (78)
Positive	54/245 (22)
Adjuvant Treatment, n (%)	
Negative	153/221 (69.2)
Positive	68/221 (30.8)
Recurrence Status	
Recurred	83/218 (38.1)
Did not recur	135/218 (61.9)
Death Status	
Alive	142/251 (56.6)
Dead	109/251 (43.4)
Median time to recurrence (years)	2.78
Median time to death (years)	3.19

**Table 2 ijms-25-00725-t002:** Patient characteristics of subcohort who received neoadjuvant therapy.

	Patients Who Received Neoadjuvant Therapy
Total, n	54
Sex, n (%)	
Male	37/54 (68.5)
Female	17/54 (31.5)
Mean age, yrs	66.41
pT category, n (%)	
T1–2	16/54 (29.6)
T3–4	38/54 (70.4)
pN category, n (%)	
N0	28/54 (51.9)
N1–2	26/54 (48.1)
pM category, n (%)	
M0	52/53 (98.1)
M1	1/53 (1.9)
Grade, n (%)	
1–2	50/54 (92.6)
3	4/54 (7.4)
Vascular invasion, n (%)	
Absent	46/54 (85.2)
Present	8/54 (14.8)
Perineural invasion, n (%)	
Absent	40/54 (74.1)
Present	14/54 (25.9)
Adjuvant Treatment, n (%)	
Negative	21/44 (47.7)
Positive	23/44 (52.3)
Tumor Regression Grade, n (%)	
0–1(Good response)	6/51 (11.8)
2–3 (Poor response)	45/51 (88.2)
Recurrence Status	
Recurred	24/52 (46.2)
Did not recur	28/52 (53.8)
Death Status	
Alive	32/54 (59.3)
Dead	22/54 (40.7)
Median time to recurrence (years)	2.25
Median time to death (years)	3.08

**Table 3 ijms-25-00725-t003:** p21 expression in different tissue types.

Tissue Type	p21 Expression		Total
	Positive	Negative	
Tumor Center (TC), n (%)	66/265 (24.9)	199/265 (75.1)	265
Tumor Periphery (TP), n (%)	92/263 (35)	171/263 (65)	263
Normal Tissue Close to Tumor (NCT), n (%)	156/248 (62.9)	92/248 (37.1)	248
Normal Tissue Away from Tumor (NAT), n (%)	193/256 (75.4)	63/256 (24.6)	256
Lymph nodes with metastases, n (%)	21/115 (18.3)	94/115 (81.7)	115
Adenoma, n (%)	46/58 (79.3)	12/58 (20.7)	58
Hyperplastic polyp, n (%)	40/56 (71.4)	16/56 (28.6)	56
All Tissue Types	614/1261 (48.7)	647/1261 (51.3)	1261

**Table 4 ijms-25-00725-t004:** Associations between p21 expression and clinicopathological data in all patients.

	Tumor Center (TC)		*p*	Tumor Periphery (TP)		*p*
	Negative	Positive		Negative	Positive	
	n (%)	n (%)		n (%)	n (%)	
Sex						
Male	133 (75.6)	43 (24.4)	0.802	119 (68.4)	55 (31.6)	0.109
Female	66 (74.2)	23 (25.8)		52 (58.4)	37 (41.6)	
Age						
≤70 yrs	88 (71.5)	35 (28.5)	0.214	80 (66.1)	41 (33.9)	0.731
>70 yrs	111 (78.2)	31 (21.8)		91 (64.1)	51 (35.9)	
pT category						
T1–2	61 (70.1)	26 (29.9)	0.138	58 (67.4)	28 (32.6)	0.620
T3–4	135 (78.5)	37 (21.5)		110 (64.3)	61 (35.7)	
pN category						
N0	98 (71)	40 (29)	**0.046**	90 (65.2)	48 (34.8)	0.995
N1–3	98 (81.7)	22 (18.3)		77 (65.3)	41 (34.7)	
pM category						
M0	165 (74.3)	57 (25.7)	0.200	148 (67.3)	72 (32.7)	0.476
M1	15 (88.2)	2 (11.8)		10 (58.8)	7 (41.2)	
Grade						
1–2	184 (75.1)	61 (24.9)	0.992	157 (64.6)	86 (35.4)	0.627
3	15 (75)	5 (25)		14 (70)	6 (30)	
Vascular invasion						
Absent	146 (73)	54 (27)	0.082	132 (66.7)	66 (33.3)	0.437
Present	52 (83.9)	10 (16.1)		38 (61.3)	24 (38.7)	
Perineural invasion						
Absent	166 (75.8)	53 (24.2)	0.847	144 (66.4)	73 (33.6)	0.458
Present	32 (74.4)	11 (25.6)		26 (60.5)	17 (39.5)	
Recurrence Status						
Did not recur	94 (69.6)	41 (30.4)	**0.029**	92 (69.7)	40 (30.3)	0.478
Recur	68 (82.9)	14 (17.1)		54 (65.1)	29 (34.9)	

**Table 5 ijms-25-00725-t005:** Univariate analysis of p21 expression and all variables with overall survival and disease-free survival.

	Overall Survival		Disease-Free Survival	
	HR (95% CI)	*p*	HR (95% CI)	*p*
p21, tumor center (TC)				
Positive vs. negative	0.955 (0.606–1.506)	0.843	0.623 (0.325–1.193)	0.153
p21, tumor periphery (TP)				
Positive vs. negative	1.946 (1.313–2.885)	0.001	1.450 (0.870–2.419)	0.154
Sex				
Male vs. female	1.057 (0.711–1.572)	0.784	1.098 (0.658–1.832)	0.720
Age				
<70 years vs. ≥70 years	1.382 (0.930–2.052)	0.109	0.740 (0.456–1.201)	0.223
pT category				
T3–4 vs. T1–2	1.756 (1.133–2.720)	0.012	2.687 (1.463–4.935)	0.001
pN category				
N1–2 vs. N0	1.441 (0.980–2.120)	0.064	1.302 (0.803–2.117)	0.283
pM category				
M1 vs. M0	5.096 (2.700–9.618)	<0.0001	-	-
Grade				
3–4 vs. 1–2	1.525 (0.817–2.849)	0.185	2.256 (1.030–4.942)	0.042
Vascular invasion				
Presence vs. absence	1.968 (1.301–2.979)	0.001	1.783 (1.015–3.133)	0.044
Perineural invasion				
Presence vs. absence	2.409 (1.550–3.746)	<0.0001	3.342 (1.930–5.785)	<0.0001
Neoadjuvant treatment				
Yes vs. no	0.944 (0.589–1.512)	0.810	1.815 (1.084–3.039)	0.023
Adjuvant treatment				
Yes vs. no	0.506 (0.301–0.850)	0.010	1.124 (0.661–1.910)	0.666

**Table 6 ijms-25-00725-t006:** Multivariable analysis of p21 expression with overall survival.

	Overall Survival Period	
	HR (95% CI)	*p*
p21, tumor periphery (TP)		
Positive vs. negative	2.068 (1.290–3.316)	0.003
pT category		
T3–4 vs. T1–2	1.366 (0.816–2.286)	0.235
pM category		
M1 vs. M0	3.444 (1.422–8.342)	0.006
Vascular invasion		
Presence vs. absence	1.687 (0.950–2.997)	0.074
Perineural invasion		
Presence vs. absence	2.184 (1.194–3.996)	0.011
Adjuvant treatment		
Yes vs. no	0.346 (0.196–0.613)	<0.0001
	Overall Survival	
	HR (95% CI)	*p*
p21, tumor center (TC)		
Positive vs. negative	0.888 (0.503–1.566)	0.682
pT category		
T3–4 vs. T1–2	1.576 (0.945–2.628)	0.081
pM category		
M1 vs. M0	3.617 (1.517–8.621)	0.004
Vascular invasion		
Presence vs. absence	1.752 (0.999–3.072)	0.050
Perineural invasion		
Presence vs. absence	1.951 (1.075–3.541)	0.028
Adjuvant treatment		
Yes vs. no	0.332 (0.186–0.590)	<0.0001
	Disease-Free Survival	
	HR (95% CI)	*p*
p21, tumor periphery (TP)		
Positive vs. negative	1.769 (1.027–3.049)	0.040
pT category		
T3–4 vs. T1–2	1.841 (0.970–3.494)	0.062
Grade		
3–4 vs. 1–2	1.952 (0.819–4.648)	0.131
Vascular invasion		
Presence vs. absence	1.370 (0.713–2.633)	0.344
Perineural invasion		
Presence vs. absence	2.465 (1.312–4.631)	0.005
Neoadjuvant treatment		
Yes vs. no	1.816 (1.034–3.191)	0.038
	Disease-Free Survival	
	HR (95% CI)	*p*
p21, tumor center (TC)		
Positive vs. negative	1.422 (0.737–2.745)	0.294
pT category		
T3–4 vs. T1–2	2.039 (1.079–3.853)	0.028
Grade		
3–4 vs. 1–2	1.727 (0.735–4.060)	0.210
Vascular invasion		
Presence vs. absence	1.369 (0.714–2.626)	0.344
Perineural invasion		
Presence vs. absence	2.182 (1.188–4.006)	0.012
Neoadjuvant treatment		
Yes vs. no	1.875 (1.062–3.311)	0.030

**Table 7 ijms-25-00725-t007:** p21 expression comparison with neoadjuvant treatment in LARC patients.

Neoadjuvant Therapy	Tumor Center		*p*	Tumor Periphery		*p*
	Negative, n (%)	Positive, n (%)		Negative, n (%)	Positive, n (%)	
Yes	40 (74.1)	14 (25.9)	0.954	36 (67.9)	17 (32.1)	0.551
No	140 (73.7)	50 (26.3)		120 (63.5)	69 (36.5)	

**Table 8 ijms-25-00725-t008:** Tumor Regression Grade (TRG) classification.

Grade	Description
0	No viable cancer cells
1	Moderate response Single or small groups of tumor cells
2	Minimal response Residual cancer outgrown by fibrosis
3	Minimal or no tumor cells killed

## Data Availability

The data presented in this study are available on request from the corresponding author. The data are not publicly available due to data size and privacy.

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
