# Peer review of "p21 as a Predictor and Prognostic Indicator of Clinical Outcome in Rectal Cancer Patients"

_ijms, 2024, doi:10.3390/ijms25020725_

Round 1

Reviewer 1 Report

Comments and Suggestions for Authors

The article “p21 as a predictor and prognostic indicator of clinical outcome in rectal cancer patients” investigated the utility of p21 as a biomarker to identify rectal cancer patients suitable for neoadjuvant and adjuvant therapy. This study found that negative p21 expression in TP cells was closely associated with better overall survival outcomes. In general, this manuscript is well organized and written and I would recommend the publication of the manuscript after the following issues are addressed:

1. It is necessary to define positive and negative p21 expression. What expression level is defined as “positive” and what is “negative”?

2. By comparing the overall survival of patients with p21-positive and p21-negative expression in tumor center tissues and tumor peripheral tissues, only negative p21 expression in TP tissues showed significance. Where is this difference between tumor center tissues and tumor peripheral tissues from?

3. It is described in the manuscript “However, upon modifying for confounders in multivariable analysis, negative p21 expression in TP samples correlated with longer DFS, independent of perineural invasion or being recipients of neoadjuvant treatment (p21, TP-available cases [HR = 1.769 (1.027-3.049), p= 0.040]).” However, this key conclusion was not supported or reflected in Figure 2.

4. Please cite some relevant works in the manuscript “Supramolecular Polymerization-Induced Nanoassemblies for Self-Augmented Cascade Chemotherapy and Chemodynamic Therapy of Tumor. Angew. Chem. Int. Ed. 2021, 60, 17570–17578.”

Reviewer 2 Report

Comments and Suggestions for Authors

Comments to the authors:

The manuscript presents a study on the relationship between p21 expression in rectal cancer cells and patient survival outcomes. While the study contributes valuable insights, several aspects could be critically addressed:

1: Limited Contextualization of p21 Expression: The text mentions altered p21 expression in rectal cancer cells without providing a comprehensive background on the role of p21 in the cell cycle, cellular senescence, and cancer development. A more extensive introduction could enhance the understanding of the significance of p21 in the context of rectal cancer.

2: Study Design and Methodology: The text briefly mentions using tissue microarrays and immunohistochemical staining for p21 in rectal cancer specimens. However, it lacks details on the study design, sample selection criteria, and potential biases. A more thorough description of the methodology would strengthen the study's credibility.

3: Statistical Analysis: While the study presents statistical analyses, it would be beneficial to include information about the statistical methods employed, including the choice of tests and adjustments for multiple comparisons. Transparency in statistical approaches enhances the reproducibility and reliability of the study.

4: Patient Demographics and Characteristics: The text provides basic demographic information, but a more detailed analysis of patient characteristics, such as comorbidities, lifestyle factors, and socioeconomic status, could contribute to a more comprehensive understanding of the study population.

5: Incomplete Data and Small Subcohort: The text acknowledges limitations related to incomplete data on treatment regimens and causes of death. The small size of the subcohort that received neoadjuvant therapy is also highlighted. Discussing the potential impact of these limitations on the study's findings would provide a more realistic interpretation.

6: Comparison with Previous Studies: The text mentions conflicting results with previous studies regarding the correlation between p21 expression and clinical outcomes. However, a deeper exploration and discussion of these discrepancies, including potential reasons for differences, would add depth to interpreting the study's findings.

7: Clinical Relevance and Implications: While the study identifies an association between negative p21 expression in tumor periphery cells and better survival outcomes, it is crucial to discuss the clinical relevance of these findings and their potential implications for patient management and treatment decisions.

8: citations: the following paper should be cited in the revised version. Overcoming MTDH and MTDH-SND1 complex: driver and potential therapeutic target of cancer, CI 2024, 3 (1)29https://doi.org/10.58567/ci03010004

9: Clarity in Presentation: The text includes substantial data and results, which could be better organized and presented for clarity. Graphical representations of critical findings, such as survival curves, could enhance the reader's understanding.

Incorporating these critical considerations could strengthen the overall quality and impact of the study.

Comments on the Quality of English Language

 Minor editing of English language required
